# Recent Developments in Heteroatom/Metal-Doped Carbon Dot-Based Image-Guided Photodynamic Therapy for Cancer

**DOI:** 10.3390/pharmaceutics14091869

**Published:** 2022-09-05

**Authors:** Rajkumar Sekar, Nagaraj Basavegowda, Saktishree Jena, Santhoshkumar Jayakodi, Pandian Elumalai, Amballa Chaitanyakumar, Prathap Somu, Kwang-Hyun Baek

**Affiliations:** 1Department of Chemistry, Karpaga Vinayaga College of Engineering and Technology, GST Road, Chengalpattu 603 308, Tamil Nadu, India; 2School of Biotechnology, Yeungnam University, Gyeongsan 38541, Korea; 3Department of Biotechnology, Karpaga Vinayaga College of Engineering and Technology, GST Road, Chengalpattu 603 308, Tamil Nadu, India; 4Department of Biotechnology, Saveetha Institute of Medical and Technical Sciences (SIMATS), Saveetha School of Engineering, Chennai 602 105, Tamil Nadu, India; 5Department of Biotechnology, University Institute of Engineering and Technology, Guru Nanak University, Hyderabad 500 085, Telangana, India

**Keywords:** carbon dots, photodynamic therapy, cancer theranostics, bioimaging

## Abstract

Carbon nanodots (CNDs) are advanced nanomaterials with a size of 2–10 nm and are considered zero-dimensional carbonaceous materials. CNDs have received great attention in the area of cancer theranostics. The majority of review articles have shown the improvement of CNDs for use in cancer therapy and bioimaging applications. However, there is a minimal number of consolidated studies on the currently developed doped CNDs that are used in various ways in cancer therapies. Hence, in this review, we discuss the current developments in different types of heteroatom elements/metal ion-doped CNDs along with their preparations, physicochemical and biological properties, multimodal-imaging, and emerging applications in image-guided photodynamic therapies for cancer.

## 1. Introduction

Cancer is a significant civic health problem, accounting for almost 10 million deaths per annum globally [1,2,3]. The existing traditional methods for cancers are surgery and chemo/radiotherapy, which have numerous side effects and poor biological barriers [4,5,6]. For this reason, huge efforts are currently being made to enhance modern approaches with both highly specific therapeutic efficacy and minimal side effects in healthy cells. The near-infrared (NIR)-based photo-ablation therapies comprising photodynamic therapy (PDT) and photothermal therapy (PTT) are emerging as important cancer therapies because of their non-invasive character combined with less cytotoxicity, multi-drug resistance, and minimum harm to healthy cells. Moreover, phototherapies can act together with other modal therapies, namely chemo-, gene-, and immunotherapy, to reduce side effects and develop a supreme way of healing for patients [7,8,9,10,11,12]. Figure 1 exhibits the mechanisms of photodynamic therapy once the photosensitizers (PSs) have been treated for NIR irradiation with a suitable wavelength range. Initially, electrons are excited from the S_0_ (ground state) to the excited S_1_ (singlet state) by the absorption of NIR, and then follow the three pathways: (i) return from the S_1_ to S_0_ via non-radioactive emission in the form of heat, (ii) return from the S_1_ to S_0_ via radioactive emission, in between the surface strap states present due to surface impurities, defects, and functional groups, and the sub-sequential development of fluorescence, and (iii) conversion starting from the S_1_ to the excited T_1_ triplet form via intersystem crossing (ISC). Finally, from the T_1_ to S_0_ ground state, free radicals are generated via electron transformation to create reactive oxygen species (ROS) [13]. The increasing level of ISC can also increase T_1_, which is useful for sensitizing O_2_ to provide ^1^O_2_ and develops the PDT effect [14]. To date, many organic PSs have been reported, namely Phthalocyanine [15], hypocrellin [16,17], and porphyrin [18], which have been utilized in PDT for cancer. Moreover, the reported organic PSs have been restricted due to water insolubility, aggregation, less specificity to tumor cells, and side effects in healthy cells [19,20,21]. Along with organic PSs, metal and semiconductor-based nanomaterials have been reportedly used in PDT with strong absorbance and expressed high photothermal transfer efficiency. The heavy cytotoxicity of metal ions restricts their usage in clinical trials [22,23,24]. As a result, the growth of new nanomaterials is necessary for use in PDT for cancer treatment.

Carbon nanodots (CNDs) are the new members of the carbon family along with carbon-based nanotubes, diamonds, graphenes, and fibers. CNDs are the latest nanostructure of 0D carbon having a nanosize of 2–10 nm with marvelous luminescent properties [25]. CNDs established with sp2 hybridized carbon form the core and are surrounded by many functional groups, namely –COOH, NH, OH, C-N, SH, and an aggregation of polymers. CNDs were first discovered accidentally at the time of the refinement of carbon nanotubes by electrophoresis by Xu et al. in 2004 [26]. CNDs have received great interest in the area of biomedical science due to their preparation, quick surface modification ability, unique optical and physicochemical properties, low cytotoxicity, and excitation wavelength of fluorescence and photoluminescence emission [27]. Due to these special characteristics, they have revealed their importance in the area of bio-imaging and biosensors. Moreover, CNDs have the capability to transform NIR energy into ROS and thus can be applied as nanocarriers or PSs in PDT [28,29,30,31].

In recent years, many materials scientists have been extensively employing various heteroatoms (fluorine (F), boron (B), phosphorus (P), nitrogen (N), and sulfur (S)) in addition to metals (Ga, Mg, Cu, Zn, Ni, Co, Ca, Au, Fe, etc.) as doping materials to improve the physicochemical properties of CNDs [32,33,34]. The attachment of metals can modify the electronic structure of CNDs, thereby changing the HOMO–LUMO energy gap on which the optical characteristics of CNDs rest. The luminescent property of CNDs can also be improved because of the surface plasmonic resonance (SPR) nature of metals [35]. In the case of the metal-doped carbon nanodot M-CNDs, surface plasmonic resonance (SPR) plays a potential role along with the carbon core and the presence of emissive trap states in the surface of M-CNDs. The energy transfer in the M-CNDs causes non-radiative recombination, resulting in enhanced fluorescence quenching. The SPR differs inversely with the ionic size of the metal; the smaller ionic-sized metal ions exhibited higher fluorescence intensity. M-CNDs displayed a surprisingly efficient lifetime due to the SPR-coupled photoluminescence. In heteroatoms, the nitrogen doping of CNDs is the most appreciable strategy for enhancing the CNDs’ properties [36]. Consequently, sulfur (S) doping was also established to be beneficial for enhancing the optical properties of CNDs [37]. Other dopants, namely boron (B) [38] and phosphorus (P) [39] have been also used. The development of doped CNDs with heteroatoms/metal ions improves their NIR-induced ROS production rate due to the hopeful photoluminescence and the photo-induced electron transfer properties of CNDs. Currently, numerous review articles have been published on both heteroatom- and metal-doped CNDs and their biomedical applications in cancer therapies. However, no review article has reported on both the metal- and heteroatom-doped CNDs discussing their applications in PDT of cancer alone so far. 

In this review, we summarize the recent reports on successful PDT and PDT-based combination therapies based on doped CND-based nanostructures through various surface alterations and fabrication of doped CNDs (Figure 2). This review offers a comprehensive overview of the physicochemical and biological properties of doped CNDs for use in PDT for cancer and how they are altered and fabricated as nanoplatforms for PDT-based cancer therapy.

## 2. Synthesis Techniques for Doped CNDs for PDT

Doping is an efficient way to alter the essential properties of CNDs. By presenting heteroatoms/metal ions doped into CNDs, their electronic structures can be altered, promoting n- or p-type carriers. Therefore, their optical and electronic characteristics are linked to the HOMO–LUMO band gaps and they are able to change by using dopants. Additionally, current progress in research has shown that doping could largely improve the quantum yield of CNDs [40]. Commonly, there are two types of methods that can be used to prepare the CNDs, namely top-down and bottom-up methods. Recent methodologies for the synthesis of doped CNDs are summarized in Table 1. In the top-down process, heavy carbon resources (graphite, coal, ash) are decomposed into CNDs, whereas in the bottom-up process, organic matter is converted and polymerized into CNDs [25]. 

In the top-down process, sulfuric and nitric (strong) acids are employed to oxidize the carbon bulk materials so the surface of CNDs is surrounded by carboxyl and hydroxyl groups [41,42,43]. The amount of these oxygen-comprising functional groups can be quickly controlled by tuning the pH range of the acids. Moreover, it is very hard to dope heteroatoms/metal ions into CNDs. Additionally, using strong acids may destroy the π-conjugated forms in the CNDs, as it creates low absorption and low-emission wavelengths. Thus, the top-down process is not adaptable for the large-scale preparation of CNDs. However, in the bottom-up process, CNDs are synthesized using the solvothermal method for the organic substances. Commonly, this is an eco-friendly method using non-toxic solvents and produces a high yield compared to the top-down approach; thus, it is fit for bulky synthesis. The surface functional groups on CNDs are majorly dependent on the chemical formula of carbon materials, pH value, pressure, solvents, and temperature. Different kinds of heteroatoms/metal ions can be quickly doped into CNDs [44]. Furthermore, the doping amount and sites are also hard to regularize. The photoluminescence properties of doped CNDs, particularly their emission and absorption wavelengths, are vital for their photodynamic application in cancer therapies. The doped CNDs with strong NIR fluorescence and absorption can be utilized for successful diagnoses and the destruction of tumors in deep tissue [45].

**Table 1 pharmaceutics-14-01869-t001:** Heteroatom/metal-doped CNDs along with their synthesis methods, quantum yields, and colors.

Heteroatom-Doped CNDs					
Synthesis Method	Doping Agent	Precursors	Quantum Yield (%)	Emission	Ref.
Hydrothermal	N	Folic acid	23	Blue	[46]
	F	PEI 600 Da and 2,2,3,3, 4,4-hexafluoro-1,5-pentanediol diglycidyl ether	5.6	-	[47]
	P	Sodium citrate and phytic acid	3.5	Blue	[48]
	B	Phenylboronic acid	12	-	[49]
Solvothermal	N	Carbon tertrachloride and diamines	9.8–36.3	Blue	[50]
	F	Tetrafluoroterephthalic acid	-	Green	[51]
	P	Hydroquinone and phosphorous tribromide	25.1	Blue	[39]
	B	Hydroquinone and boron tribromide	14.8	Blue	[52]
Microwave	N	citric acid-malonic acid-oxalic acid-succinic acid	90	Blue	[53]
	P	Ethylenediamine/phytic acid	21.65	Green	[54]
	B	Citric acid, urea, and boric acid	15	Green	[55]
**Metal-Doped CNDs**					
Hydrothermal	Cu	Poly(methacrylic acid) and Cupric nitrate	80	Blue	[56]
	Zn	Glucose oxidase and glucose, zinc chloride	32.3	Blue	[57]
	Mg, N	Citric acid and Magnesium hydroxide	83	Blue	[58]
	Cu, N	citric acid monohydrate, copper acetate monohydrate	50.1	Blue	[59]
Solvo thermal	Zn	Citric acid monohydrate, urea, zinc chloride	51.2	Yellow	[60]
	Mn	Ethylenediaminetetraacetic acid and manganese chloride tetrahydrate	90.79	Blue	[61]
Micro wave	Fe, N	L-Tartaric acid, urea, FeCl_3_.6H_2_O, oleic acid	-	Blue	[62]
	Gd	Diethylene glycol (DEG), sucrose, and Gd_2_O_3_	5.4	Green	[63]

## 3. Physicochemical and Biological Properties of Doped CNDs

CNDs as 0D carbon group materials have excellent biocompatibility and their lateral dimensions are typically less than 10 nm [64]. The preparation techniques for doped CNDs are entirely dependent on the approach used (top-down or bottom-up), which have been systemically described [64,65]. Apart from these old-style methods for preparing doped CNDs using carbon sources, currently, many eco-friendly green preparation tactics have been utilized. Well-organized heteroatom/metal-doped CNDs with good hydrophilic properties were prepared by an eco-friendly approach using natural polysaccharides as a precursor [66]. In addition, well-formed heteroatom/metal-doped CNDs have been synthesized with exceptional photoluminescence (optical) and biocompatibility (biological) properties. Furthermore, various advanced techniques in the characterization process have provided a strong underpinning for the usage of doped CNDs in the field of biomedical science.

### 3.1. Physicochemical Properties of Heteroatom/Metal-Doped CNDs

Most reports confirm that heteroatom/metal doping of CNDs could disturb the pristine electronic structure of CNDs. Thus, the PL properties of doped CNDs are linked to the HOMO–LUMO band gaps that have been changed due to the different types of heteroatom/metal-doped CNDs. The high quantum yield in the doped CNDs enhances the PL properties and also introduces some novel physicochemical properties, namely the magnetic resonance imaging (MRI) relativity rate and catalytic performances. In this section, these physicochemical properties will be exposed and debated.

#### 3.1.1. Photoluminescence in CNDs

The most popular and attractive property of CNDs is its optical (PL) properties since its clear mechanism is still unpredictable. Many proposed mechanisms for CNDs’ PL are based on their surface effect, quantum confinement effect, and molecule-like state. In CNDs, a smaller particle size leads to quantum confinement, and many researchers have shown that the energy/band gap of the carbon core energy level increases with the decreasing size of CNDs, subsequently in an emission wavelength with the blue shift of CNDs [67,68]. However, the PL properties arise from the surface effect of the CNDs and develop an improved quantum yield, decay lifetime, and emission wavelength. Many functional groups on CNDs’ surface can develop various surface effect energies, which can offer emissive energy levels and create several PL colors [69,70]. Many reported CNDs show blue and green PL emission wavelengths in the range of 400–500 nm; moreover, some red PL emissions have also been observed [71,72,73]. It is important to note that CNDs with good quantum yields and lifetimes are required in cancer theranostics, particularly to enhance image-guided photodynamic therapy [74].

#### 3.1.2. Photoluminescence in Heteroatom/Metal-Doped CNDs

Heteroatom/metal-doped CNDs showed improved PL properties due to the surface defects via the increasing number of energy traps. Due to this effect, the energy gap of CNDs decreases and the non-radioactive electron transfer has less of an impact and consequently leads to the enhancement of the quantum yield of CNDs [37,70]. In heteroatom doping, the N atom is predominantly used as a dopant and causes a red shift in the emission wavelength of CNDs and an increase in the quantum yield. It can be allocated to the surface energy states created by doping the N onto CNDs, developing a radioactive reunion and reducing the chance of the non-radioactive reunion of excited electrons. Moreover, electron-releasing functional groups such as the amino groups can also increase the excited energy states’ stability by improving the pi electrons’ conjugation system in the CNDs skeleton. This can improve the electron transition from the ground state to the excited state so the N atom can contribute to the quantum yield of CNDs.

Nowadays, the S atom is also employed as a dopant to improve the quantum yield of CNDs [75,76]. Zhang et al. [77] investigated the electronic structures of carbon dots (CDs), S-doped CDs, and N, S co-doped CDs to describe the high quantum yield of the red and orange emissions of N, S co-doped CDs. The observed results show that the heteroatoms can create more electronic states and in-gap trap states as recombination points. The heteroatoms (N and S) can quickly bind to carbon owing to their similar atomic radii and the accessibility of the 5 and 6 electrons in their valence band. Thus, N and S doping can alter the band gap of CDs and increase the probability of electronic transition from the singlet ground level to the triplet excited level, concluding in a remarkably high quantum yield of CDs. Similar to the above investigation, Dong et al. [74] analyzed the outcome of N and S co-doped CDs and demonstrated a noteworthy improvement in the quantum yield of S,N co-doped CDs of 73% compared to N-doped CDs with 16.9% and CDs with 5.3%. Moreover, the observed lifetime of N and S co-doped CDs (12.11 ns) was higher than that of non-doped CDs (7.45 ns). The results confirmed that a novel surface energy state formed in CDs through the N-doping method, together with the further co-doping of the S element, can remarkably improve the density of this state. Certainly, the S element can remove the O elements from the surface state and become stable and then improve the surface state of the N element. The optical properties of the doped CNDs are summarized in Table 1. 

In recent years, many metal ions have been doped to enhance the luminescence properties of CNDs. The metal doping agents can enhance the quantum yield due to the presence of the valence electron and its electron-transferring processes, which aid the radioactive recombination of holes and electrons on the externally modified surface of the CNDs [32,78,79]. Generally, metal-doped CNDs exhibit a solid and broad absorption band between 200 nm and 600 nm. Compared with CNDs, the emission color of metal-doped CNDs is greatly improved in the visible space at a similar level of concentration owing to the possibility of a charge transfer between the metal dopants and graphite [80,81]. Moreover, most of the metal-doped CNDs show emissions with various colors of fluorescence in a 360 nm UV-vis region, such as yellow [82], green [63], blue [83], blue-green [84], and so on. When compared with CNDs, metal-doped CNDs exhibit a red shift in the fluorescence emission owing to their excitation-based luminescence properties and the emission peaks also demonstrate a higher wavelength with the intensity slowdown as the excitation wavelength exhibits a red shift [56,61,85,86,87,88]. Moreover, almost all oxygen-holding functional ligands on the metal-doped CNDs decrease owing to their chemical treatment with the doping metal ions but still show high solubility in aqueous and organic solvents. For instance, Wang et al. [89] showed that the emission fluorescence color of Mn-doped carbon dots changed from blue to yellow as the solvent polarity increased, whereas the emission color of the non-doped CDs was blue with similar solvents and denotes the vital role of Mn doping on CNDs. Heteroatom/metal-doped CNDs can synergistically deliver disease imaging and targeting via PDT, which can be utilized as a new approach for forthcoming cancer theranostics.

### 3.2. Biological Properties of Doped CNDs

Even though the possibilities of using CNDs in nanomedicine are vast, it is essential to study the vital biohazards of CNDs due to the interactions between the CNDs’ biomaterials and the body’s organs [90,91,92]. Basically, once CNDs are offered for a particular biomedical use, they must elevate their adequate bio-functions, which can lead to clean renal clearance without side effects [93,94,95]. Therefore, it is vital to systematically evaluate the toxic nature of heteroatom/metal-doped CNDs from the perspective of toxicology studies. To determine the cytotoxicity in vitro, a few biological chemical pointers, namely apoptosis, DNA damage, and reduced oxidative stress, are commonly utilized to fully demonstrate the biocompatibility of heteroatom/metal-doped CNDs with a cell-line medium. Various intravenous processes to introduce CNDs can effectively alter the physiological system and disturb the pharmacokinetics, which depends on the timeline and outcome of the intake, circulation in the bloodstream, metabolic activity, bio-distribution, renal excretion, and bio-interface interaction of superficially introduced CNDs in an organism [96,97,98]. 

The quantum yield improvement of doped CNDs is noticeable and important due to the heteroatom/metal doping, which sorts doped CNDs into more convenient biomedical applications. Moreover, the related issue of cytotoxicity caused by the doping of heteroatoms/metals should be observed particularly for their vital role in cancer theranostics applications. Evaluating the cytotoxicity of CNDs through in vitro and in vivo assays to allow for their biomedical applications is necessary for their supplementary advancement. To date, many scientists have assessed the in vitro biocompatibility of different heteroatom/metal-doped CNDs. Commonly, in vitro cytotoxicity models expose cell viability measurements via specific assays namely CCK-8, MTT, and WST-1. Due to their nanosize, which is a major factor affecting their toxicities, CNDs indicate fairly small cytotoxicity in many in vitro models. 

Surface-modified CNDs with functional groups, namely –NH_2_, –OH, –COOH, and heteroatom/metal-doped CNDs, also show low cytotoxicity. Table 2 shows a summary of the biocompatibility experiments of heteroatom/metal-doped CNDs. Many reports have demonstrated the outstanding cell viability of heteroatom-doped CNDs [99,100,101,102]. For instance, Edison et al. [103] showed that the nitrogen-doped carbon dots have low cytotoxicity on HeLa cells. Xu et al. [104] achieved a 100% cell viability assay with HeLa and HepG2 cells on doped carbon dots after 24 and 48 h. The biocompatibility reports showed that metal-doped CNDs had low toxicity [83,84,105,106,107,108,109,110,111]. For instance, Xu et al. [112] placed an in vivo investigation of Gd-doped carbon dots in liquid form (20 mmol/kg) into mice via a tail vein. After a week of observation, the histological differences in the heart, liver, spleen, kidney, and intestine were analyzed through hematoxylin and eosin staining. In this study, the results showed that there were no significant variations between the control and experiment groups and there were no tissue pathological impairments resulting from the intravenous injection of Gd-doped carbon dots.

## 4. Multimodal-Imaging of Doped CNDs

The fluorescent nature of CNDs is very adaptable for direct application in optical imaging. Owing to the occurrence of various emissive levels, CNDs exhibit excitation-based PL. This excitation-based PL property can be utilized to obtain multicolored images of cancer cells. CNDs exhibiting different color emissions with doped elements have been used for fluorescence imaging for PDT. Fluorescence imaging (FI) includes various imaging methods namely, magnetic resonance imaging (MRI), computed X-ray tomography (CT scan), and photoacoustic (PA) imaging, which have different ranges of imaging depths, spatial resolutions, and sensitivities, and all of them are generally applied for the earlier diagnosis of cancer and pre-clinical proposes (Figure 3) Multimodal bio-imaging can combine the benefits of individual imaging modalities, which results in more accurate statistics for therapy design and for guiding therapeutic models [119]. With the aim of enhancing the sensitivity and accuracy of cancer diagnoses, the synthesis of the doped CNDs with multimodal bio-imaging capabilities is highly recommended. General multimodal bio-imaging of CNDs comprises MR/FI, PA/FI, and CT scan/FI.

### 4.1. MR/FI of Doped CNDs

MR imaging is a non-invasive technique for the early diagnosis and monitoring of diseases during therapy with a high potential for spatial resolution for infected soft tissues. The combination of MR and FI should become an influential non-invasive technique with high spatial resolution and is intensely anticipated [120,121,122]. In recent years, there have been three classes of MR-based contrast agents such as the paramagnetic metal ligands complex, gadolinium (Gd)- or Mn (T_1_)-based complex, and nanoparticle (iron oxide) T_2_-based contrast agents. The CND-based MR/FI bimodal imaging system can be delivered by the doping of Gd or Mn metal ions into CNDs. Recently, Gd-established complexes have been frequently hybridized with CNDs to generate nanocarriers. In addition, Gd^3+^ metal ions can be openly doped into CNDs using doping techniques to form Gd-doped CNDs for MR/FI multimodal bio-imaging [123]. Zou et al. synthesized Gd-CDs via the hydrothermal carbonization of gadopentetic acid and glycine. The Gd-CDs exhibited encouraging biocompatibility with good performance at a T_1_ relaxivity rate of 6.45 mM^−1^ s^−1^ and radio sensitization properties, which could be applied to the MR/FI bimodal imaging-guided radio treatment of cancer. Consequently, Gd-doped carbon dots can be used in MR/FI bimodal bio-imaging and decrease the cytotoxicity as well as enhance the longitudinal relativity rate [124].

Wang et al. [125] prepared a Gd-based contrast agent Gd@C_82_, which shows a better MRI contrast property than the commercially available Magnevist. The longitudinal relaxivity rate of Gd@C_82_ and Magnevist in water are 27 mM^−1^ s^−1^ and 3.5 mM^−1^ s^−1^ at 7.1 T, respectively. Subsequently, after surface modifications with the -OH/NH_2_ group, Gd@C_82_ can generate a blue light emission under photoexcitation. Moreover, this group of modified Gd@C_82_ with red emission carbon dots and PEG can enhance the dual-modal bio-imaging and PDT. Wang and Zhang et al. [126] developed boron (B)-doped carbon dots as an alternative T_1_ bright agent for use in MRI. The magnetization value of B-CDs was comparatively lower than the well-established Gd-based agents, which offers a new pathway for application in bimodal bio-imaging. On the other hand, Mn-doped CNDs have also been applied for MR imaging. Wang et al. [127] developed Mn-doped CDs using a hydrothermal process with manganese (II) phthalocyanine. Later, after self-assembly with DSPE-PEG, the formed doped CDs could be employed as a contrast agent for a combined T_1_-weighted MR (relaxivity 6.97 mM^−1^ s^−1^)/FI. Gu et al. [128] developed Mn-doped CDs for MR/FI multimodal analysis for blood–brain glioma bio-imaging. Mn-doped CDs exhibited photo-excitation-based emissions and the Mn3+ metal ions delivered the CDs as a successful T_1_-weighed MR imaging agent.

Further, the Mn-CDs could develop an improved MR contrast effect in the more complex blood–brain glioma sites, which could potentially make them useful in MR/FI multimodal bio-imaging probes to locate blood-brain gliomas. Recently, Han et al. [61] investigated the T_1_-weighed MR contrast imaging of Mn-CDs by inoculating intravenously into HO-8910 cancerous nude mice. From Mn-CDs, the T_1_ MR-contrast images were developed at various time intervals. Pre- and post-inoculation results were compared with Mn-CDs on T_1_-weighed signals in the cancer region and improved by 2.06, 2.33, and 2.52 times at 5, 30, and 60 min, respectively. These simultaneous imaging outlines confirmed the gathering of the Mn-CDs in the tumor sites, signifying their importance as a novel type of tumor targeting T_1_ MRI contrast agents. Moreover, images of the kidneys could be observed, a sign of the renal clearance of the inoculated Mn-CDs. Luo et al. [129] developed novel liposomal-based carbon dots with iron (Fe@CDs) through self-assemblies of amphiphilic lipopeptides (Figure 4). In this study, Fe@CDs showed good photothermal and gene transfection effects in cancer cells. Furthermore, doped carbon dots could also function as a multimodal imaging agent for PA/PT/MR/FI, exhibiting an order-of-magnitude improvement in the PA signals and 16 °C thermal increases after a 5 min laser treatment on cancer sites.

### 4.2. PA/FI of Doped CNDs

Photoacoustic imaging (PAI) is a booming non-invasive imaging technique that can offer an optimized optical contrast and deep imaging of an infected region. When a pulsed laser beam is treated on the nanomaterials it produces isochoric thermal energy, which generates a thermo-elastic explosion as well as ultrasound (US) waves, also named photoacoustic (PA) waves. The PA contrast imaging technique is able to combine optical sensitivity and acoustic depth organ permeation for early disease diagnosis with improved accuracy. Due to their potential photothermal transformation properties, doped CNDs can be utilized as exogenous contrast imaging agents to enhance the PA waves [130,131,132,133,134]. Recently, NIR-triggered cancer theranostics can be applied, which include photodynamics with real-time photodiagnostics, namely PA and FI, which have been dynamically followed due to their spatiotemporal targeting and specificity for cancer treatment. For instance, Wang et al. [131] synthesized sulfur-doped CDs (SCDs) with red PL by using polythiophene phenylpropionic acid as a precursor. The SCDs can concurrently behave in FI, PA, and photothermal ways for cancer imaging and therapy in pre-clinical mice experiments. The SCDs exhibited a good photothermal conversion proficiency of 38.5%, and the thermal range of the aqueous dispersions increased to 26.6 °C with the accumulation of the SCDs (200 µg/L). The SCDs can gather in organs, such as the kidneys, lungs, and liver, and retain fairly constant PA motion after an extended time circulating in the blood stream during an all-inclusive diagnosis.

The biodistribution of the SCDs in the cancer sites showed improved and flawless PA imaging of cancer regions in mice during pre-clinical studies. Co-doped phosphorus (P) into nitrogen-doped carbon dots (P, N-CDs) can potentially lead to a localized energy state nearer to the Fermi region, thus showing improved PA imaging. The imaging property of P, N-CDs was investigated for in vivo PA and FI using cancerous nude mice as the animal model [133]. Recently, Tian et al. [135] effectively fabricated nickel and nitrogen co-doped carbon dots (Ni-CDs) using a hydrothermal method for imaging-guided phototherapy. The Ni-CDs showed noteworthy absorption in the second near-infrared area with a notable photothermal conversion efficiency of as high as 76.1% under 1064 nm with a power density of 0.5 W cm^−2^ and also functioned as PTAs for multimodal PA/MR/PTI-guided photothermal therapy in the second near-infrared area. In this study, in vivo PA and MR imaging of U14 cancerous mice before and after 12 h of an intravenous injection of Ni-CDs were investigated and exhibited thermal increases in the cancerous region of the U14 cancerous mice in the control as well as Ni-CDs with the second near-infrared group under a 1064 nm laser treatment at specific intervals of time. 

### 4.3. CT Scan/FI of Doped CNDs

In addition to the other bio-imaging systems (MRI/FI and PA/FI), computed tomography scanning (CT-scan) is a type of non-invasive imaging used in the clinical diagnosis of diseases [136]. In recent years, Su et al. [137] have fabricated ruthenium-doped carbon dots (Hf-CDs) from thiourea, citric acid, and ruthenium chloride. The prepared Hf-CDs can be successfully applied for the in vivo CT-scan/FI multimodal bio-imaging of orthotopic liver cancer with tumor targeting and active renal clearance. In this study, various organs from mice including kidney, lung, spleen, and liver were forfeited at various durations such as 1, 10, 30, and 60 min post-injection to investigate the distribution and renal clearance of Hf-CDs. Consequently, prepared doped CDs have good targeting capability and noticeable CT-scan/FI processes. Zhang et al. [138] prepared iodine (I)-doped carbon dots (I-CDs) from iodixanol and glycine using a hydrothermal method. The prepared I-CDs exhibited PL emissions of 475 nm and superior X-ray CT. Furthermore, in vivo studies showed that I-CDs were bio-distributed on the infected sites and applied for X-ray CT imaging as well as clean renal clearance. Zhao et al. [139] reported that a novel variety of gadolinium and ytterbium-doped carbon dots were prepared using the hydrothermal method for the multimodal imaging of MR/CT/FI. In this study, the prepared Gd/Yb@CDs exhibited a better CT scan and T_1_ MR imaging at (45.43 HU/L g^−1^) and (r1 = 6.65 mM^−1^ s^−1^), respectively. The Gd/Yb@CDs showed optical properties by excitation of the dependent emissions as shown in Figure 5. A summary of the multimodal system of CNDs is presented in Table 3.

## 5. Image-Guided Doped CNDs for PDT

PDT has attracted huge attention in basic research, as well as pre-clinical trials, owing to its non-invasive nature, high efficiency against drug resistance, minimal side effects, and minimal damage to the peripheral tissues than currently used cancer therapies. [146,147,148,149]. In PDT, the PSs gather in malignant cancer cells and, following laser treatment of PSs by a suitable wavelength, generate ROS from intracellular O2, which initiates apoptosis in cancer cells. [150,151]. Thus PSs and photo excitation play vital roles in enhancing PDT. The combination of PSs with doped CNDs can elevate the intracellular uptakes by changing the diffusion mechanism due to their high water solubility, superficial surface modification, good photo stability, biocompatibility, and photon absorption characteristics. Particularly, heteroatom/metal-doped CNDs a with unique nature play a vital role in obtaining ideal PDT cancer theranostic nanomaterials. Moreover, doped CNDs as nanocarriers can realistically be used in image-guided PDT. In this section, we discuss the recently doped CND-based nanocarriers that have emerged for use in PDT, which are dependent on heteroatom- and metal-ion-doped CND nanomaterials as PDT nanoagents.

### 5.1. Image-Guided Metal-Ion-Doped CNDs for PDT

Hu et al. [152] developed innovative and very stable Sn atom-doped carbon dots, which guarantee the effective creation of ^1^O_2_ (ROS) and improve fluorescence and the PDT effect. In this study, Sn@CDs showed a high quantum yield of 58.3% (^1^O_2_) with low cytotoxicity on 4T1 cancer cells both in vitro and in vivo, as well as good photoluminescence and water solubility. Furthermore, on irradiation of LED of 400–700 nm with a power density of 40 mW/cm^−2^, a 25% reduction in 4T1 cancer cells with a small impact on healthy cells was observed. Sn@CDs showed that they could reduce tumors and act as good PS, showing a great potential for PDT application. Irmania et al. [153] synthesized amine-functionalized manganese-doped carbon quantum dots (Mn-CQDs). Consequently, folic acid (FA) and chlorin e6 (Ce6) were conjugated to form the Mn-CQDs@FA/Ce6 as a magneto-red photoluminescent property for PDT applications. Mn-CQDs@FA/Ce6 exhibit red PL characteristics at pH 6, 7, and 8 from 0 to 72 h and 7T MRI with a different range of nanomaterial concentrations both in longitudinal (1/*T*_1_) and transversal (1/*T*_2_) relaxation rates.

In this study, dual-modal imaging and therapy were carried out using Mn-CQDs@FA/Ce6. Their exposed relativity value was 5.77 and they showed a red PL at 360 nm and provided more photodynamic cytotoxicity to HeLa cancer cells via folate-mediated endocytosis. The origin of ROS in cancer cells is vital to the success of PDT. The tumor microenvironment initiating the ROS by Cu- and N-doped carbon dots in mouse melanoma cancer cells suitable for NIR laser treatment with 808 nm utilizing 2′,7′-dichlorofluorescin diacetate (DCFH-DA) as a tumor cell penetrable non-emissive agent was investigated. In this experiment, ROS were oxidized by DCFH-DA with outstanding specificity to a deep-green photoluminescent dichlorofluorescein (DCF). The appearance of the emission of green luminescence within mice cancer cells showed the production of ^1^O_2_ by doped CDs laser treated with 808 nm NIR for 10 min. From these results, the prepared Cu,N-CDs can successfully create ^1^O_2_ when treated by NIR light in the tumor microenvironment, which can oxidize DCFH-DA, so doped CDs can be used to develop a new photodynamic sensitizer agent. The preparation of a new model of Cu,N-CDs with NIR absorption fitting for combined in vivo PDT and PTT practices is shown in Figure 6 [154].

Zheng et al. [155] developed CDs as nanocarriers incorporating cisplatin Pt(IV)-DOX with imageguided therapy against the drug resistance of cisplatin in usual cancer treatment. The successful cellular uptake of the drugs DOX and Pt(IV) was followed by the release of the drugs from their CD nanocarriers under a cancerous cell microenvironment, and then finally, the Pt(IV) and DOX ran their mechanisms to offer a combined therapeutic efficiency and initiated the successful cell death of A2780 and A2780 cells, which were resilient to Pt(IV). The successful tumor-specific PDT was achieved by constructing nanocarriers to improve oxygenation in the cancer cells of microregions where acidosis and hypoxia, as well as raised levels of H_2_O_2_, are important. For example, Chen et al. [156], developed CDs/manganese dioxide (CDs/MnO_2_) nanocomposites surface-functionalized with PEG to create hydrophilic CDs/MnO_2_-PEG nanocomposites. 

In this study, the developed nanomaterials exhibited fluorescence emission, good ^1^O_2_ generation, and improved MR imaging in cancer cells. In in vitro studies on HeLa cells, the results showed that the acidic environment of cancer cells gave red fluorescence at a 200 g mL^−1^ nanomaterial concentration with a 635 nm NIR treatment and induced enhanced apoptosis. On other hand, in vivo studies on BALC mice exhibited the effective relief of cancer hypoxia due to the MnO_2_-stimulated breakdown of hydrogen peroxide into molecular oxygen. This way of improving oxygenation in the tumor microenvironment was a better condition for enhancing the efficiency of image-guided in vivo PDT. Thus, the in vitro and in vivo experiments indicated that the CDs/MnO_2_-PEG nanocomposites could be used as a pH/H_2_O_2_-driven, turn-on type of theranostics for multimodal FL/MRI and oxygen-increased PDT for solid cancer therapies. This was also demonstrated by Ge et al. [127] who utilized Mn(II) phthalocyanine to prepare Mn-doped CDs using the solvent thermal method. ROS production by Mn(II)-doped CDs was oxidized via H_2_O_2_ addition, although with the low concentration of oxygen, Mn-doped CDs could generate ROS and molecular oxygen creators to progressively create ^1^O_2_. Consequently, this route relocated the triplet energy of PSs to O_2_ to form highly reactive ^1^O_2_.

### 5.2. Image-Guided Heteroatom-Doped CNDs for PDT

Owing to their excellent photoelectron transfer behavior, doped CNDs can develop ROS via photoelectron transfer from doped CNDs to molecular oxygen. This advantage can be utilized in PDT, which uses ROS for cancer therapy. In heteroatom-doped CNDs, the ROS generation by nitrogendoped CDs synthesized via the solvothermal transformation of coal resources was investigated, and an ^1^O_2_ generation with good quantum yield (19.0%) was obtained, which denotes their great importance in PDT [157]. Nitrogen-doped carbon dots (NCDs) exhibited both targeting ability and cytotoxicity in tumor cells, showing remarkable inhibition of tumor growths under 630 nm laser irradia-tion in in vivo anticancer therapy. Further, chlorin-e6-conjugated NCDs showed enhanced fluorescence under 430 nm excitation, and a PDT effect on MGC803 cells in the concentra-tion range of 0–50 μM [158]. Zhao et al. [159] developed S and N atom co-doped carbon dots (S,N-CDs) using a hydrothermal method, which resulted in photon fluorescence imaging as well as synergistic photothermal and photodynamic therapies for cancer. In this study, the developed multi-functional heteroatom-doped carbon dots showed that the co-doped (S and N) carbon dots had higher therapeutic efficiency than single doping (N only), and the ^1^O_2_-producing ability and photothermal conversion of S,N-CDs was 27%, and 34.4%, respectively. Xu et al. [142] designed selenium- and nitrogendoped carbon dots (Se/N-CDs) as a photosensitizer using RNA as a carrier. In this study, RNA plays a vital role and acts as a carrier and transports the Se/N-CDs close to the nucleus, consequently breaking the nuclear membrane under laser treatment. The prepared Se/N-CDs generated only 10.6% of the ^1^O_2_, which is comparatively lower than commercially available photosensitizers.

## 6. Conclusions and Outlook

Heteroatom/metaldoped CNDs as a new member of photoluminescent carbon-family-dependent nanomaterials have excellent potential for use in cancer theranostics owing to their unique physicochemical properties. The fluorescence emission from the UV-visible to near-infrared regions provides the CNDs with the ability to bio-image guided cancer treatment, and thus lead to a pathway for the real-time footprint of bio-distributed drugs on cancer sites. Additionally, heteroatom/metal-doped CNDs have excellent biocompatibility, which reflects the multimodal non-invasive MR/CT/PA imaging potential of doped CNDs. In this review article, we discussed the current progress and improvements in research in the field of imaging-guided therapy for photodynamic therapy based on heteroatom/metal-doped CNDs, as there were no previous review articles that discussed this subject. This review focused on the outline of the synthesis methodology, physicochemical properties, biocompatibility, multimodal imaging, and finally, the image-guided PDT applications of the heteroatom/metal-doped CNDs. Doped CNDs showed marvelous physicochemical behavior and good ROS production ability and bio-imaging properties, which have often been described as being potential cancer theranostic materials for PDT. The hypoxic nature of tumor delays the molecular oxygen (O_2_)-based PDT method, and the veins of the tumor are demolished after PDT, consequently reducing the O_2_ concentration in the tumor site. Moreover, the high hypoxic region may initiate the metastasis of cancer cells. Utilizing the microenvironment sites of the tumor, namely, the hydrogen per oxide, GSH concentrations, and acidic nature of the tumor, leads to an increase in the O_2_ in situ, which is vital to clearing the hypoxia problems and enhancing the efficiency of PDT. The physicochemical properties of heteroatom/metal-doped CNDs are linked to their design, composites, and surface interface, and the existing preparation methods for doping CNDs are of comparatively low yield and commonly use natural materials as the source. Moreover, the probable cytotoxicity of heteroatom/metal-doped CNDs reduces their potential uses in clinical trials. The main methods for decreasing the side effects in healthy cells include decreasing the cytotoxicity and proper renal clearance rate of doped CNDs by tuning their size and enhancing the tumortargeting ability. Currently, one of the most basic problems is the lack of a scalable preparation producer to generate high quality doped CNDs with desirable nanostructures (for instance, size, morphology, number of functional groups, and location of defects). Therefore, for the industryscale production of doped CNDs with high performance, the effects of the metal precursors and reaction conditions on the performance of doped CNDs should be systematically explored, and a purification method dependent on size and polarity should also be developed. The enhancement of the PDT enactment of doped CNDs by nano-engineering their unique properties is still a challenging task for materials researchers. Materials scientists should expand the potentially efficient microwave-assisted process owing to their environmentally friendly nature and fast reaction kinetics. Metal-doped CNDs have mostly been used for MR/FI imaging purposes. Therefore, other types of multimodal imaging in the area of Ct scan/FI and PA/FI using exact in vivo models are required. Finally, more effort is needed in the area of image-guided photodynamic therapy using heteroatom/metaldoped CNDs. Even though doped CNDs have had various encounters in pre-clinical trials in PDT due to progress in nanomedicine, the above difficulties will be slowly overcome and it is projected that they will be used for early diagnosis and personalized nanomedicinebased cancer therapy.

## Figures and Tables

**Figure 1 pharmaceutics-14-01869-f001:**
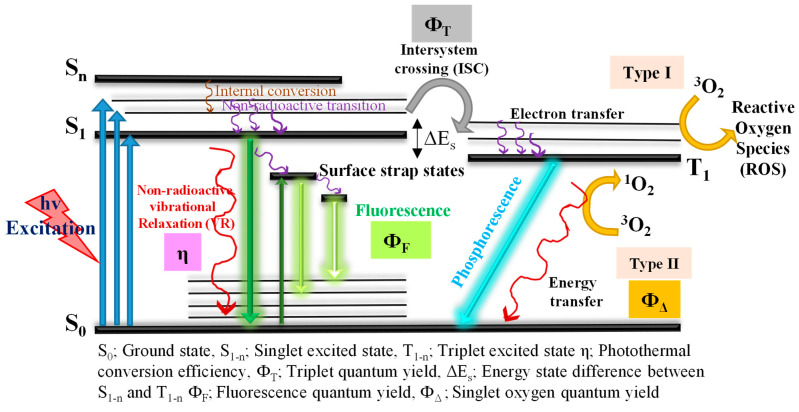
The mechanisms of photodynamic therapy.

**Figure 2 pharmaceutics-14-01869-f002:**
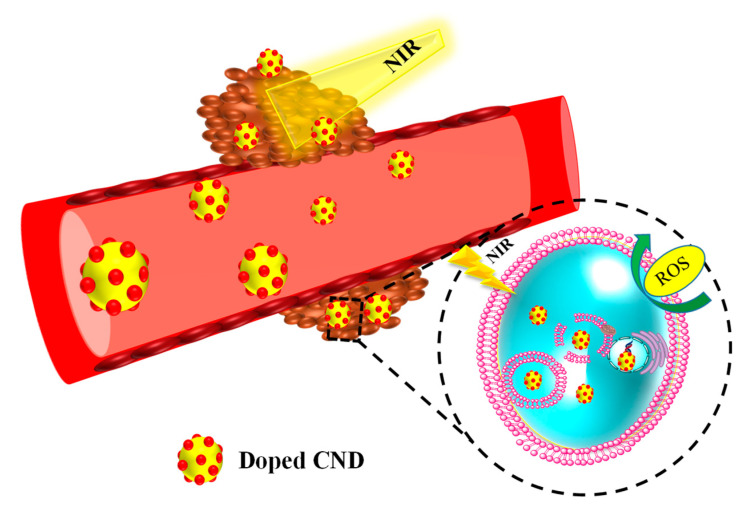
Doped CND-based nanostructures for PDT for cancer.

**Figure 3 pharmaceutics-14-01869-f003:**
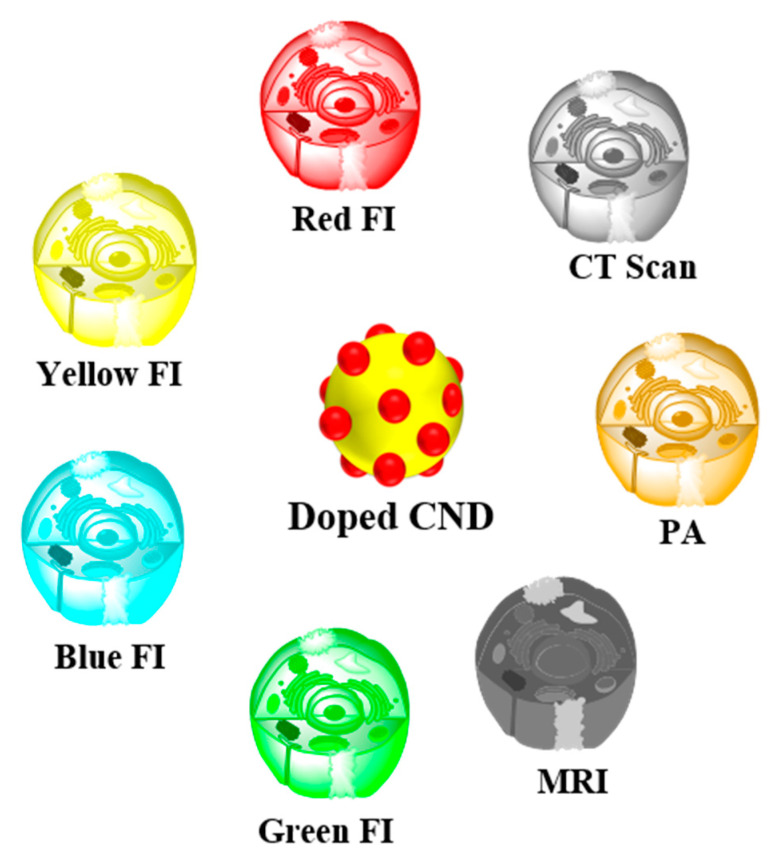
Various imaging techniques of heteroatom/metal-doped CNDs.

**Figure 4 pharmaceutics-14-01869-f004:**
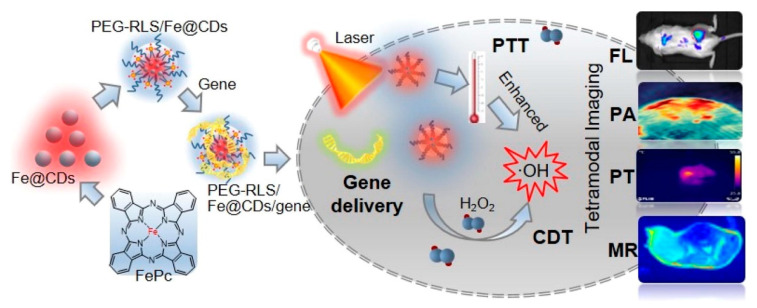
Graphical representation of the iron-doped carbon dots as a theranostic agent for PTT, gene delivery, and multimodal (MR/PA/PT/FI) imaging-guided PTT/chemodynamic therapies. Reproduced with permission [129]. Copyright 2021, Elsevier.

**Figure 5 pharmaceutics-14-01869-f005:**
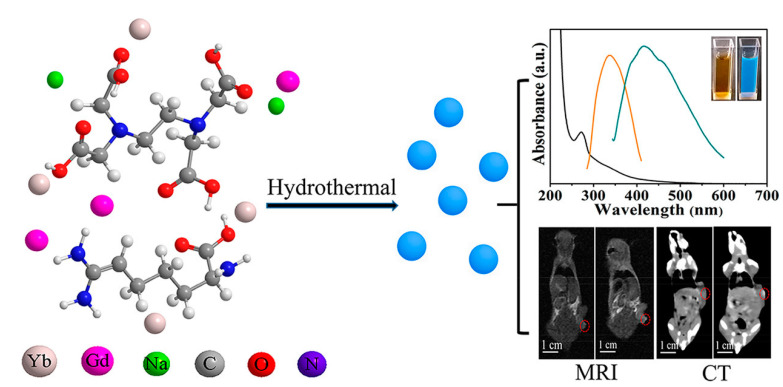
Schematic representation of preparation of Gd/Yb@CDs, UV-visible spectrum of doped carbon dots with optical metaphors under 365 nm UV and daylight, and finally, MR and CT images of cancer region observed pre- and post-intravenous administration for 5 h and 24 h, respectively. Reproduced with permission [139]. Copyright 2018, ACS.

**Figure 6 pharmaceutics-14-01869-f006:**
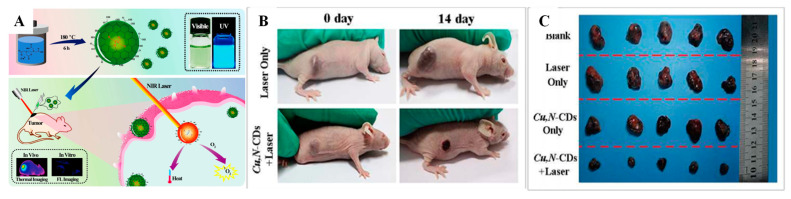
(**A**) Graphical representation of Cu,N-CDs with multimodal theranostics of combined PTT/PDT of mouse melanoma (B16) cells. (**B**) Photographic images of tumor-bearing mice with laser treatment on different days. (**C**) Corresponding photographs of tumors in the blank group (Blank without treatment), control groups (only Cu,N-CDs without laser and Laser only applied without Cu,N-CDs), and test group (Cu,N-CDs + Laser). In test group, significant reduction in the tumor size observed, while both control test group (Cu,N-CDs and Laser only) no effect is observed in the tumor size in comparison with blank group. Reproduced with permission [154]. Copyright 2018, Elsevier.

**Table 2 pharmaceutics-14-01869-t002:** Biocompatibility of doped CNDs.

S. No.	Doped-CND	Cell Model	Assay	Incubation	Viability/ Concentration	Ref.
1.	Mg–EDA–CDs	L929	MTT	24 h	90% (250 mg mL^−1^)	[58]
2.	Mn-CDs@Anti-HE4	HO8910	MTS	24 h	85% (3 mg mL^−1^)	[61]
3.	Gd-CDs	C6	MTT	24 h	83% (1 mg mL^−1^)	[63]
4.	Gd-QCDs	NIH3T3	MTT	24 h	121.4 mg mL^−1^	[113]
5.	NPCDs	HepG2	MTT	24 h	88% (100 mg mL^−1^)	[114]
6.	Te-CDs	HeLa	MTT	24 h	80% (200 mg mL^−1^)	[96]
7.	N-O-CDs	HeLa	MTT	24 h	80%	[115]
8.	S, Se-codoped CDs	HeLa	MTT	24 h	>80% (40 mg mL^−1^)	[116]
9.	PMn@Cdots/HA	HEL	WST-1	24 h	100% (20 mg mL^−1^)	[117]
10.	MnNS:CDs@HA	B16F1	WST-1	24 h	90% (500 mg mL^−1^)	[118]

**Table 3 pharmaceutics-14-01869-t003:** Multimodal imaging properties of heteroatom/metal CNDs.

S. No	Doped CND Materials	Diagnosis Modes	Therapies Applied	Ref.
1.	AS1411-Gd-CDs	MR/FI	PTT	[140]
2.	Dox@IR825@Gd@CDs	MR/FI	CHEMO/PTT	[141]
3.	Se/N-CDs	MR/FI	PTT/PDT	[142]
4.	GNR@SiO_2_-CDs	PA/FI	PTT/PDT	[143]
5.	Gd/Yb@CDs	MR/CT/FI	-	[139]
6.	I-CQDs-C225	CT/FI	-	[144]
7.	Ce6-RCDs	PA/FI	PTT/PDT	[145]
8.	Mn-CDs@Anti-HE4	MR/FI	-	[61]

## Data Availability

Not applicable.

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
