# Peer review of "Recent Developments in Heteroatom/Metal-Doped Carbon Dot-Based Image-Guided Photodynamic Therapy for Cancer"

_pharmaceutics, 2022, doi:10.3390/pharmaceutics14091869_

Round 1

Reviewer 1 Report

In my opinion, the work presented is very impressive. The review is based on 159 articles from the literature in recent years. It summarises the knowledge on the new generation of the carbon-based nanomaterial family and their application in anti-cancer therapy. In particular, the authors, in my opinion, brilliantly discussed the current developments of different types of CNDs doped with metal heteroatoms/ions, together with their repair, physicochemical and biological properties, multimodal imaging and their emerging applications in image-guided photodynamic therapy for cancer.

My main research field concerns the synthesis of carbon-based nanomaterials, which is why I found this review extremely interesting and comprehensive, as it brings together recent developments on this fascinating family of nanomaterials. Furthermore, the conclusions are consistent with the topics presented. For these reasons, I recommend the publication of the manuscript in its present form.

Author Response

  Manuscript ID: pharmaceutics-1867065

 Dear Editor and reviewers

Thank you very much for your kind consideration and invaluable efforts to review our manuscript. We are truly grateful for the meaningful comments and insights of the reviewers and editor. We have carefully revised all issues raised by the reviewers. We feel that our manuscript has been significantly improved from the reviewer’s creative and insightful suggestions. We would like to submit our fully revised manuscript after revised of all comments. All corrections made are highlighted in green color in the revised manuscript. Please see the response to reviewers’ comments on the succeeding pages.

Thank you very much for your consideration.

Reviewer 1

In my opinion, the work presented is very impressive. The review is based on 159 articles from the literature in recent years. It summarises the knowledge on the new generation of the carbon-based nanomaterial family and their application in anti-cancer therapy. In particular, the authors, in my opinion, brilliantly discussed the current developments of different types of CNDs doped with metal heteroatoms/ions, together with their repair, physicochemical and biological properties, multimodal imaging and their emerging applications in image-guided photodynamic therapy for cancer.

My main research field concerns the synthesis of carbon-based nanomaterials, which is why I found this review extremely interesting and comprehensive, as it brings together recent developments on this fascinating family of nanomaterials. Furthermore, the conclusions are consistent with the topics presented. For these reasons, I recommend the publication of the manuscript in its present form.

Comment 1

English language and style are fine/minor spell check required

Response

We are thankful for the reviewer's note. As suggested by the reviewer, we have checked the minor spell check and English revisions.

Reviewer 2 Report

This review article carefully summarized the synthesis and photodynamic therapy of N, S/metal doped carbon dots. Several aspects of image-guided therapy and brief limitations of doped carbon dots are included. Under this concern and the comments below, I suggest this paper could be published in pharmaceutics after minor revision.

1. At the center of figure 1, there are two states (marked as short thick black line) which can give fluorescence emission in the figure. What are those two states? The authors have carefully explained the ground state, singlet excited states and triplet excited states. It will be helpful for the readers in pharmaceutics if the authors could make these two states clear and explain how electrons transfer from the S1 state to these two states.

2. In line 78, the authors stated that “The luminescence property of CND can also improve because of surface plasmonic resonance (SPR) nature of metals.” To my understanding, metals are usually with high electron affinity, which will trap electrons produced in the photoinduced charge separation process, thus interrupted radiative charge recombination and efficiently quenched fluorescence of carbon dots. So how metals improve the luminescence of CND?

3. In line 515, the authors talked about the “un-purified” problem of CND. It is suggested that the authors could explain slightly more on this issue to raise the attention of material scientists on contaminated samples.

Some typos:

Line 81, “Sulfur(S)” à “sulfur (S)”

Line 138, “Science” à “science”

Line 149, 155, “Man” à “Many”

Line 471, 474, “NCDs” might be a typo, I guess the author want to say “CNDs”

Author Response

Dear Editor and Reviewers
Manuscript ID: pharmaceutics-1867065
Dear Editor and reviewers
Thank you very much for your kind consideration and invaluable efforts to review our
manuscript. We are truly grateful for the meaningful comments and insights of the reviewers
and editor. We have carefully revised all issues raised by the reviewers. We feel that our
manuscript has been significantly improved from the reviewer’s creative and insightful
suggestions. We would like to submit our fully revised manuscript after revised of all comments.
All corrections made are highlighted in green color in the revised manuscript. Please see the
response to reviewers’ comments on the succeeding pages.
Thank you very much for your consideration.

Reviewer 2

This review article carefully summarized the synthesis and photodynamic therapy of N, S/metal doped carbon dots. Several aspects of image-guided therapy and brief limitations of doped carbon dots are included. Under this concern and the comments below, I suggest this paper could be published in pharmaceutics after minor revision.

Comment 1

At the center of figure 1, there are two states (marked as short thick black line) which can give fluorescence emission in the figure. What are those two states? The authors have carefully explained the ground state, singlet excited states and triplet excited states. It will be helpful for the readers in pharmaceutics if the authors could make these two states clear and explain how electrons transfer from the S1 state to these two states.

Response

We are thankful for the reviewer note. As suggested by reviewer, in figure 1 the two short lines between the S1 and S0 states are surface strap states have mentioned in the figure and explained in the revised manuscript. Kindly see below or line number 43-51in the revised manuscript

Initially, electrons are excited from the S0 (ground state) to the excited S1 (singlet state) by absorption of NIR. and then following the three consequences pathways: (i) return from S1 to S0 via non-radioactive emission in form of heat (ii) then return from S1 to S0 via radioactive emission, in between these surface strap states are present due to surface impurities, defects, and functional groups, sub-sequential development of fluorescences and (iii) to conversion starting S1 to the excited T1 triplet form via intersystem crossing (ISC). Finally, from T1 to S0 ground state, generates free radicals via electron transformation and to create reactive oxygen species (ROS) [13].

Comment 2

In line 78, the authors stated that “The luminescence property of CND can also improve because of surface plasmonic resonance (SPR) nature of metals.” To my understanding, metals are usually with high electron affinity, which will trap electrons produced in the photoinduced charge separation process, thus interrupted radiative charge recombination and efficiently quenched fluorescence of carbon dots. So how metals improve the luminescence of CND?

Response

We are grateful to the reviewer. Yes, I agreed with the point of reviewer. We explained how metals improve the luminescence of CND. Kindly see below or line number 81-87 in the revised manuscript.

 In case the metal doped carbon nanodots M-CNDs, surface plasmonic resonance (SPR) comes into potential role along with the carbon core and the presence of emissive trap states in the surface of M-CNDs. The energy transfer in the M-CNDs roots non radiative recombination resulting in enhanced fluorescence quenching. The SPR differs inversely with ionic size of the metal, the smaller ionic size metal ions   exhibited higher fluorescence intensity. M-CNDs displayed efficiently exciting lifetime due to SPR coupled photoluminescence. 

Comment 3

In line 515, the authors talked about the “un-purified” problem of CND. It is suggested that the authors could explain slightly more on this issue to raise the attention of material scientists on contaminated samples.

Response

We are highly grateful to the reviewer for highlighting this important point. As suggested by reviewer, “the issue of un-purified CND” was more briefly explained in the revised manuscript. Kindly see below or line number 524-530 in the revised manuscript

Currently, one of the most basic and potential problem is the lack of a scalable preparation producer to generate high quality doped CNDs with desirable nanostructures (for instance, size, morphology, numbers of functional groups, and location of defects). Therefore, for industry scale production of doped CNDs with high performance through an potential route, effects of metal precursors and reaction conditions on the performance of doped CNDs should be systematically explored, and a purification method depend on size and polarity also adequate to be developed.

Comment 4

Some typos:

Line 81, “Sulfur(S)” à “sulfur (S)”

Line 138, “Science” à “science”

Line 149, 155, “Man” à “Many”

Line 471, 474, “NCDs” might be a typo, I guess the author want to say “CNDs”

Response

As suggested by reviewer, we have corrected the above spelling mistakes in the revised manuscript. Kindly see below in the revised manuscript

Line 89, “sulfur (S)”

Line 147, “science”

Line 158, 164, “Many”

Line 480, 484, “NCDs” NCDs refers nitrogen doped carbon dots as per the references.

Nitrogen doped carbon dots (NCDs) exhibited both targeting ability and cytotoxicity in tumor cells, showing remarkable inhibition of tumor growth under 630 nm laser irradia-tion in in vivo anticancer therapy. Further, chlorin-e6-conjugated NCDs showed enhanced fluorescence under 430 nm excitation, and a PDT effect on MGC803 cells in the concentra-tion range of 0–50 μM [158].

Comment 5

English language and style are fine/minor spell check required

Response

We are thankful for the reviewer's note. As suggested by the reviewer, we have checked the minor spell check and English revisions.
